# Generation Z and Key-Factors on E-Commerce: A Study on the Portuguese Tourism Sector

**Jorge Vieira \*** , **Rui Frade, Raquel Ascenso, Inês Prates and Filipa Martinho**

Department of Management and Administration, ISLA Santarém, Higher Institute of Management and Administration, Largo Cândido dos Reis, 2000-241 Santarém, Portugal; rui.frade@islasantarem.pt (R.F.); raquel.ascenso@islasantarem.pt (R.A.); inesprates96@outlook.pt (I.P.); filipa.martinho@islasantarem.pt (F.M.)
\* Correspondence: vieira.jm@gmail.com; Tel.: +351-243-305-880

**Abstract:** In recent years, tourism has experienced remarkable growth worldwide. This sector is rapidly becoming the main export activity and the most important source of GDP growth in several countries. In Portugal, it represented around 19.7% of exports in 2019. The internet and online platforms contributed decisively to this growth. Generation Z already represents a considerable portion of society and, in the coming years, will become the central consumer segment. With this research, we intend to identify the key factors in the decision to purchase online, in Generation Z individuals, in the Portuguese tourism sector. We carried out a characterization of the sector, a bibliographic review and the identification of key variables. We applied a structured questionnaire to a sample of 233 individuals aged between 10 and 25 years. Subsequently, the data were processed using descriptive methodologies and association tests between variables. The key factors in the decision to purchase tourism products/services in Generation Z are Trust, Price, the use of aggregating websites, WOM/EWOM, the Offer of products/services online and the Online experience. In the opposite direction, it was given less importance to Convenience, Reviews on tourism websites, Tourism Advertising, Social networking and the possibility of Canceling the reservation. This study allows us to establish the bases for future research, to help researchers to understand Generation Z consumption habits.

**Keywords:** generation Z; e-commerce; tourism

## 1. Introduction

Generation Z already represents a considerable part of society. In 2013, Generation Z represented about 23% of the world population (Singh 2013) and, in the short term, will be the key generation of consumers. Known to be "digital natives", highly qualified, technologically experienced, innovative and creative individuals (Duffett 2017; Flippin 2017; Priporas et al. 2017), it is expected Generation Z members to be the key generation to lead the e-commerce consumer behavior in the upcoming years (Monaco 2018).

The tourism sector was the largest export activity in Portugal in 2019, accounting for 19.7% of total exports and with a contribution of 8.7% to the Portuguese GDP (PORDATA 2019). This sector has experienced significant growth with the intensification of the use of the internet and e-commerce platforms. According to Marktest (), about 192,000 residents in Portugal over the age of 15 (22.4% of internet users), claim to have purchased at least one tourist product online. Individuals with more online shopping habits, belong to higher social classes, are predominantly middle/senior managers and are aged between 25 and 34 years. According to the same study, in the last 6 years, there was a 34% growth in the number of online consumers of tourism products/services.

With this research, we intend to identify the determining factors for the acquisition of tourism products by individuals of Generation Z. We intend to anticipate the predominant buying behaviors in e-commerce. This generation will represent the most relevant group of consumers in electronic commerce, in the upcoming years. We decided to study the purchase of tourism products/services due to the growing importance of electronic commerce in the growth of this sector. Bearing in mind the impact of the tourism sector revenues to the countries' economies and, in particular, to Portugal's GDP, we expect that the results of this study may be of great relevance for tourism operators and marketers in general.

## 2. Theoretical Background

### 2.1. Generation Z

According to Étienne et al. (2008), segregating the population into generations aims to characterize a group of individuals who were born in the same historical period and have similar cultural and social backgrounds, resulting in communalities between their perceptions, interests and behaviors. However, all generations suffer changes depending on the context of each historical period. Each generation cannot be defined exclusively by a sample of individuals of the same age group, but also by a set of values, concepts and lifestyles, which they share in common (Silva 2017).

It seems there is a consensus in international literature that allows for the identification of five different generations. However, the limits that separate each generation are not entirely coincident between authors: Silent Generation (from 1930 to 1945), Baby Boomers (from 1946 to 1964), Generation X (from 1965 to 1977), Generation Y (from 1978 to 1994), Generation Z (from 1995 to 2009) and Generation Alpha (from 2010 onwards) (Chaney et al. 2017; Williams and Page 2011).

The letter Z in Generation Z comes from the expression *zapping*, characterizing this generation after an expression used to describe the act of rapidly changing TV channels in search for topics of interest, ignoring everything else (Berkup 2014). It refers to the generation of individuals born between the mid-90s up until 2010, often characterized by "screen addicts". Generation Z individuals do not know a world without continuous and permanent access to the internet (Duffett 2017). Generation Z is also referred to as net-Generation, e-Generation or iGen and its individuals often called *homo sapiens digitalis*, digital natives or post-Millennials. This generation was born under the influence of the advent of new technologies, smartphones, tablets, Wi-Fi, online gaming and social networks, setting them apart from previous generations such as the Baby Boomers (Barclays 2013; Meirinhos 2015). Most of these individuals will not be able to remember nor understand the world without smartphones or social media. However, Generation Z individuals already make up around 23% of the U.S. population (Statista 2020). In Portugal, Generation Z individuals represented 15% of total population in 2019. The oldest members will be turning 25 this year (United Nations 2019).

Generation Z can be individualized by a set of features. They are characterized mainly for their complete trust in technologies, open-mindeness, intelligence, enthusiasm, innovative and entrepreneurial spirit, as well as for being defenders of ethical and deontological principles. These individuals are highly qualified, technologically experienced, innovative and creative individuals (Flippin 2017; Priporas et al. 2017).

Toledo et al. (2012) add that this generation is highly critical. They frequently change their opinion, are very concerned with environmental issues and are demanding, versatile and flexible professionals. Generation Z was born into a deep technological world and can't live without having access to the internet or to technological devices, through which they share information, express their opinions, consumption desires and feelings (Gollo et al. 2019; Vizcaya-Moreno and Pérez-Cañaveras 2020).

Bejtkovský (2016) identifies some other characteristics of Generation Z: they are less open to receive different opinions and find it difficult to cope with interpersonal relationships, except in a virtual environment or social networking. When compared to other generations, Generation Z individuals seems to have relationship issues, often preferring to communicate through online tools.

Generation Z's consumption behavior patterns allow us to distinguish them from other generations. These individuals love to discover new things on the internet and other digital platforms. They are enthusiastic when sharing their opinions and feelings about the experience of buying goods and/or services, are careful about how they spend their money on those activities, doing heavy research to identify the best options and value for their money (Posner 2015). According to Berkup (2014), Generation Z's individuals value all comments about products and/or services or ideas and/or experiences, shared on their network. According to Neto et al. (2010), these comments result in a more critical and well-formed opinion. Vizcaya-Moreno and Pérez-Cañaveras (2020) add that understanding consumer's behavior is complex, especially when it comes to a young audience, as they are increasingly demanding and live in a fast pace.

Mangles (2017) states that digital natives are no longer influenced by traditional marketing techniques and are less sensitive to traditional media. According to Törőcsik et al. (2014), marketing plays an important role in focusing strategies on these consumers, since they represent a considerable percentage of the population and have an increasing purchasing power. This generation's online research behavior serves exclusively to find the information needed to better support purchase decisions, look for online recommendations and to make reservations (Monaco 2018). According to Tutek et al. (2015), this generation requires all their expectations to be met by companies, in particular, what concerns to having reliable, in real-time information, and to have the possibility of interacting and contributing by creating and sharing their experiences in different media.

Tapscott (2010) named eight attributes to characterize this generation:

- Freedom: Generation Z individuals value their freedom, from freedom of choice to freedom of expression. He also mentions that this generation has the desire to choose the way they work, using technology to find new routes to develop their professional activity, other than the traditional rules of the workplace while combining business with domestic and social life.
- Customization: the author argues that this generation likes to personalize everything around them. This customization also extends to the scope of work, where these individuals prefer to follow new paths for their professional careers.
- Integrity: they seek integrity and openness, whether as consumers or employees, they make sure that company values are in accordance with their own values.
- Scrutiny: this generation considers to be normal the research for relevant information about companies and products/services, in order to get better transparency on their reputation.
- Collaboration: Generation Z is also known as the generation of collaboration and relationships. They participate online in different groups, prefer online gaming, interacting with other participants, and use the internet to share information. Through social networking, they discuss and share opinions about companies, brands, products and services.
- Entertainment: this generation values the availability of having entertainment in their workplaces, in their education or in their social life.
- Speed: because they were born in a digital environment, they value the speed of things. They are used to having instant responses and virtual conversations in real-time—this will make communication faster than ever. As consumers, these individuals prefer to have quick responses and fast deliveries when ordering a product/service.
- Innovation: they want innovative and modern products, thus contributing to their social status and a positive self-image.

Notwithstanding an increasing interest in the most recent generations, existing research on youth tourism is relatively underdeveloped. This is surprising because young tourists already have an important impact on the present and future of tourism. First, they represent an increasingly significant economic force: in 2015, almost 23% of total tourists were aged 16–29; 33% of all hotel reservations were made by young tourists while the total volume of international youth tourism is expected to double its value in the upcoming years. Compared to older generations, youth tourists are

more resilient because they don't have concerns when choosing destinations under socio-politic or environmental stress. Finally, young tourists tend to spend more on the destination than on travel and accommodation. Therefore, the young generations of travelers, such as Generation Z, can represent a major economic opportunity in general and for economically and politically fragile regions in particular (Cavagnaro et al. 2018; Monaco 2018).

Gradually Generation Z consumers will move up the population pyramid and eventually replace the older generation. If the tourism sector wants to be prepared for the future by designing future proof products and services, it has to take this generational change into account. The middle-aged tourist in 2030 will have completely different needs, wants and travel behaviors than the current tourist generation. All these considerations point to the importance of investigating this target group and the need to identify the key factors for this generation's e-consumption of tourism products/services, such as values and the meaning they give to a tourism experience (Cavagnaro et al. 2018).

Tourist operators, marketers and communication managers must adapt their strategies to this new reality. Personalize marketing strategies, services and resources, adapting them to several platforms, to better expose the value of brands and communication, thus respecting their opinions, ensuring interactivity between companies and Generation Z consumers. Monaco (2018) refers that nowadays, these individuals are the most active tourist segment on the market, compared to the other generations.

## 2.2. Internet and E-Commerce

Currently, there are about 4 billion individuals connected to the internet worldwide, 85% of which go online at least once a day. Of these, 92.6% connect through their mobile devices. Consumers spend more time with digital media and digital devices, so the accessibility to the internet, mobile technology and digital innovation has changed consumer's behavior and strongly influence the way they interact (Nielsen 2019). According to Ryan and Jones (2009), the internet is not just a new channel, it is a new type of channel, where user interaction is the key to success. It also generated changes in the way people communicate with each other, how they work, how they get information and how goods and services are purchased (Caro et al. 2011). With all this, the world became smaller, extraordinarily more dynamic and everything became just a click away (Kotler et al. 2016). Online commerce or e-Commerce is a commercial transaction carried out using electronic devices, where goods, services or information are exchanged (Salvador 2013). With several advantages when compared to traditional purchasing methods and with old security issues already solved by technological evolution, internet platforms can thus become one of the largest business channels (Lin and Rauschnabel 2016; Ryan and Jones 2009). About 94% of Portuguese people with internet access have already made at least one purchase via the internet. Tourism products/services, products related to fashion, tickets to events, books, music, stationery and technological products, are in the top of transactions made on the internet by Portuguese consumers (Nielsen 2019). Specifically, in the tourism sector, the internet is increasingly considered the most important channel for the dissemination of tourism products/services since it uses a standardized language in all countries, creating a new kind of relationship between companies and consumers, and even between tourists and tourism (Gretzel 2018).

## 2.3. Online Tourism

The notable growth of the tourism sector is linked to the economic, cultural and social development of society, having become one of the largest economic activities today, surpassing major wealth generating businesses, such as oil and the auto industries (Cunha 2013). As such, this sector is the one that contributed the most to the creation of jobs and new types of business, thus stimulating the development of micro and macroeconomics at a global level. The number of international tourists may exceed 1.8 billion by 2030, making tourism the main economic activity in the world (World Tourism Organization 2020). The internet was undoubtedly the event with the most relevant impact on tourism development (Martins 2013). Werthner and Ricci (2004) stress the importance of the internet and online commerce in this sector, mainly by increasing revenue and growth, since

everything has become simple to users, from access to information to the purchase process. In 2009, 24% of consumers who use the internet, reported having purchased plane tickets online at least once, and 17% made hotel reservations or purchased tours over the internet; in 2010 these values increased by 7% and 9%, respectively (Nielsen 2019).

## 2.4. The Consumer and the Online Purchase Process

According to Adolpho (2012), consumer habits have evolved considerably in the last 30 years. Consumers are increasingly more active, band together to talk, discuss the changes in society or simply to criticize a company, a product or a service. With the impact of new technologic developments and the growth of the internet, consumer behaviour has changed, from passive to active. A simple opinion can now become of major relevance to the companies' communication strategies. With the recent technological advances, consumers can now have the benefit of new innovative tools, thus allowing better access to companies' commercial activities, from anywhere in the world and at any time. These tools allow consumers to search for more and better information, help in their decision making and reinforce their power in commercial relations (Marques 2012). The online consumer is at the centre of the purchase decision process and is much more demanding than the traditional consumer because his opinion or a simple comment, can easily reach thousands of consumers in a short time (Fonseca 2015; Kotler et al. 2016).

Companies on the internet environment must use some traditional techniques to influence the purchase decision process and adapt the most appropriate marketing strategies, in order to create unique experiences and develop attractive products/services for their audience. Marketers must now combine some elements that go beyond the 4P's traditional view of marketing (Constantinides et al. 2010). The biggest difference between e-commerce and traditional commerce is on the level of technology and the management of the information, which is currently essential for a venture to succeed. In e-commerce, the post-sale phase is of critical importance, where the company can obtain important information about the customers' satisfaction status (Fonseca 2015).

The internet has become a strong ally of the consumers of tourism products/services because it is a useful tool to get more information about the goods you pretend to buy (Gosling et al. 2020). According to Silva and Mendes-Filho (2014), consumers read and share their comments about travel and tourist destinations online, instead of consulting the information shared by the tourism operators. With the use of internet tools, the selection and acquisition of touristic products/services, can be made with some simple and easy steps (Martins 2013).

Generation Z, the object of this study, adopts some aspects regarding the consumption of tourism products/services. They seek the best price/quality promotions, they like to enjoy life in a better way, regardless of the potential risks or dangers they may face. These individuals give their preference to the fulfilling of their wishes and relaxing experiences. However, they like to explore the full potential of destinations, such as looking for adventure activities, cultural visits, sightseeing tours, among others as well (Expedia 2017). Among the main reasons to go on holiday, this generation prefers the activities carried out during the trip, the possibility of having a unique experience, the cultural environment, the outdoor activities, always with a deep concern on the global cost of the product/service. According to Haddouche and Salomone (2018), Generation Z individuals show the desire to live a deep cultural experience, with interest and willingness to communicate with the local community, which is one of the most requested factors.

## 2.5. The Explanatory Variables

From the literature review and analysis, we realized that there are some variables that emerge in a continuous and systematic way, as explanatory dimensions of the theme of the present work, starting to theoretically frame them, so that they can serve as a basis for the quantitative study to be carried out.

Regarding the variable Offer of products and services online, according to Chatterjee and McGinnis (2010), with technological advances, namely the internet, consumers became less loyal and more demanding in

relation to the brands they consume, since there is the possibility not only purchasing products/services, but also comparing the opinions of other consumers. With all this demand from consumers and the fact that they are increasingly more informed, the offer tends to be increasingly larger and more varied (Dionísio et al. 2009). With this, consumers find it easier to buy tourism products/services (Azenha 2017).

As for the variable Convenience, according to Izquierdo-Yusta and Schultz (2011), when an individual uses a certain technology, it is necessary that the individual recognizes certain advantages, such as convenience and its usefulness in relation to the benefits that he may have when making an online purchase, such as reducing effort and time. Geraldo and Mainardes (2017), also argue that online consumers associate online commerce with convenience, adding that the reduction of time and effort in online shopping and the ease in the product/service search process are associated with the intention to online shop.

In the variable Online experience, Hernández et al. (2010) consider that the success of companies that are present online depends on their experience in the market and on the respective use of technology necessary for their presence. For Escobar and Camargo (2012), the consumer, when choosing to make an online purchase, seeks advantages that satisfy his needs, thus feeling more motivated, due to the security, ease of use of the website and the speed of delivery. The authors also argue that there are also other advantages of buying online, such as comparing prices and brands, without the presence and pressure of the seller.

Regarding Word-of-mouth (WOM) and Electronic-word-of-mouth (EWOM), for authors Litvin Stephen and Pan (2008), WOM and EWOM have a great impact for companies, with regards to their reputation, especially with the frequent use of social networks. Ladhari et al. (2011) define WOM, as traditional sources of information, to ask family and friends for an opinion on a certain product or service, using informal communication. They consider that EWOM is similar, changing only the medium where they share opinions, in this case online, through conversations or comments on social networks and/or forums.

Regarding Reviews on tourism websites, Xiang et al. (2014) considers that research is the first phase in tourism planning, with an increasing demand for opinions on this subject online, with these types of consumers investing more time in finding out about destinations, activities, products and services before making the purchase. Assis et al. (2019) argue that from the moment the consumer evaluates his online shopping experiences, he is able to form opinions and assign a certain value to it, reporting it to other consumers, through sharing on your social networks or other digital media.

In the variable Tourism advertising in tourism, according to Martins (2013), the consumer is influenced by several factors during the purchase decision process, namely social, cultural, personal, psychological factors and the power of advertising. The perception that an individual has about a given tourist destination is formed through primary sources such as experience and secondary sources such as advertising, thus realizing that advertising has enormous power in the image that consumers have before knowing a certain destination or location and that the same advertising may lead them to express purchase intention (Beerli and Martín 2004).

Regarding Social networks, according to Evans (2008), in a world that is constantly changing, new functionalities appear daily, such as the social network platforms, changing the way individuals communicate with each other Azenha (2017), making it possible for consumers to express themselves and exchange information with other people and transmit the desired information in real time, thus allowing everyone to be content promoters (Azevedo and Silva 2010).

Sousa (2019) states that websites should be easy to use, since when a consumer resorts to online commerce, he wants the information search and purchase process to be tasks that involve minimal effort, bringing advantages, such as the reduction of time. In the tourism sector, it is important that Tourism websites reference clear and straightforward search terms, functions to support purchase and help, transaction efficiency and also speed in the transmission of text and images (Kim et al. 2013).

For Ponte et al. (2015); Amaro and Duarte (2014), the ease of use of websites specialized in tourism has a significant and positive relationship in the intention to purchase tourism online.

Regarding the variable Trust, Chen et al. (2012) refer that there is no guarantee that the online seller will have some kind of opportunistic behavior, with Trust becoming a critical factor in the decision-making process of the online consumer. Consumers as a rule do not resort to companies they do not trust or have unethical or socially inappropriate behaviors to make their purchases. According to Azenha (2017), there may be consumers who wish to purchase some type of product available on the internet, but will not acquire it due to the risks. The main factor for the lack of trust is the lack of contact, in this case between consumers and companies (Cheshire et al. 2010).

In the variable Price, Fonseca (2015) refers that for many years, price was considered one of the main factors that measure the quality of products/services by the consumer. Currently, the same author says that the price is no longer so relevant for making a purchase decision. Despite this, price is still a value factor today, taking into account the age group and social class of the consumer (Martins 2013). For Tasoff and Letzler (2014), price is one of the most relevant factors in the intention to purchase tourism, with companies increasingly making promotions only in this channel, as a strategy to increase sales.

Azenha (2017), in relation to Reservation cancellation, states that the possibility of cancelling bookings is a crucial factor in the purchase of tourism products/services, in order not to create frustrations or any type of obstacles to the consumer, being important for the travel agencies to recognize the value that the consumer attributes to this factor, since they can filter tourism websites based on this component, choosing to buy tourism products/services on websites that give them the possibility to cancel with a certain time in advance instead of others.

## 3. Research Methodology and Sample

The main objective of this work was to carry out the first identification of key factors of online commerce in individuals belonging to Generation Z, specifically when purchasing tourism services in Portugal. We intend to identify the factors with greater preponderance but also the factors without meaning and/or that need to be reinforced. In order to comply with the objectives of our study, this work was divided into two parts. A first that presents a theoretical component and where the identification, review and theoretical framework of the main variables and themes of the study are carried out.

In the second part, the authors present the results of a quantitative analysis, obtained through the application of a structured survey, with which it was intended to evaluate the 11 variables identified: Offer of Online Products and Services, Convenience, Online Experience, WOM/EWOM, Reviews on Tourism Websites, Tourism Advertising, Social Networks, Tourism Websites, Trust, Price and Reservation Cancellation, in individuals belonging to Generation Z, the population object of this study. The survey also made it possible to collect information on the following variables, to better characterize the sample: age, gender, educational qualifications, district of residence and professional status.

According to Vicente et al. (2001), to conceive the sampling plan is to make a set of decisions leading to the selection of the sample. The selection in this case was made based on some concepts considered key, namely the definition of the target population, which was composed of individuals of both genders, residents in Portugal, who made at least one purchase of tourism through online mechanisms, aged between 10 and 25 years old, since this is the group covered by Generation Z. The sampling process was random, with the questionnaires being placed online, and 233 valid responses were obtained. All responses that did not meet the requirements for the universe under study were refused.

A pre-test of the questionnaire was also carried out. According to Hill and Hill (2012), a pre-test consists of a preliminary study of great utility, when the main investigation aims to confirm or extend an existing work in existing literature, with the aim of validation and selection of the most appropriate questions to be included in the final questionnaire, for use in the main investigation. In this work,

three people belonging to the universe under study were randomly interviewed. Small understanding issues were verified and resolved, thus obtaining the final questionnaire.

The application of the survey and the consequent collection of information took place from 25 May to 11 June 2020. We have carried out an analysis of the internal consistency of the questionnaire and the scales that constitute it. The results show that Cronbach's alpha for the complete questionnaire, evaluating all scales, is 0.933, which is considered a "Very Good" internal consistency, (Pestana and Gageiro 2005).

Through techniques of descriptive analysis of characterization of the sample, a succinct description of the interviewed individuals and of the variables under study was carried out, concluding about the degree of importance attributed to the different main variables that constitute the subject of the study, which allowed us to identify and characterize the determinants of Generation Z's purchase of online tourism. Finally, we use methods of correlation between the variables that describe online shopping in the tourism sector in Portugal, allowing us to identify the pairs of variables that have the highest degree of association, as well as pairs of variables that are not related to each other. Statistical analysis was performed using IBM© SPSS© software, version 25.

## 4. Research Results and Discussion

In the present framework, we have enrolled 233 individuals belonging to Generation Z, with an average age of 22 years, about 54.5% were male (see Appendix A). The distribution of respondents by the district of residence, educational qualifications and the professional situation are shown in Table 1.

**Table 1.** Distribution of respondents.

| Variable | | % |
|---|---|---|
| **District Residence** | Santarém | 40.3% |
| | Lisboon | 24.0% |
| | Leiria | 2.6% |
| | Setubal | 8.2% |
| | Aveiro | 2.6% |
| | Oporto | 9.0% |
| | Viana do Castelo | 0.9% |
| | Castelo Branco | 0.9% |
| | Faro | 1.3% |
| | Bragança | 0.4% |
| | Évora | 1.7% |
| | Coimbra | 0.9% |
| | Braga | 1.7% |
| | Guarda | 0.4% |
| | Vila Real | 0.9% |
| | Viseu | 0.4% |
| | Portalegre | 0.4% |
| | Açores | 2.1% |
| | Madeira | 1.3% |
| **Education level** | Basic education | 2.6% |
| | High school | 57.9% |
| | University education | 33.5% |
| | Postgraduate Training | 6.0% |
| **Predominant Professional Situation** | Student | 29.6% |
| | Student worker | 24.0% |
| | Self-employed | 6.4% |
| | Employed | 35.2% |
| | Unemployed | 4.7% |

Most of respondents belong to the Santarém District (40.3%), followed by the Lisbon District (24.0%). The sample consists of 57.9% of individuals with high school education, 33.5% with a

University degree, and residual values for Basic Education (2.6%) and Postgraduate Training (6.0%). Over one-third of the sample are employed, followed by students (29.6%) and student workers (24.0%). The remaining minority is divided between self-employment (6.4%) and unemployed (4.7%). The distribution curve of ages does not deviate much from symmetry and this is due to the existence of an outlier corresponding to a respondent who was 10 years old, when the majority of the sample consists on individuals between 16 and 25 years old. It turns out that the median age of women (22 years old) does not differ much from the median age of men (23 years old). About one-third of the individuals travel for tourism once a year, followed by those who travel twice a year (26.2%). These results are in accordance with some conclusions of our theoretical revision (Nielsen 2019). Many use individual travel agency websites as a research source (14.6%), but the overwhelming majority opt for aggregating websites (Booking, Momondo, etc.). A minority of individuals use social networks (8.6%) and specialized blogs/webpages specialized in tourism (5.2%).

Generation Z individuals show a tendency to buy tourism products/services online due to the available online services (M = 3.96). It is an innovative and creative generation, with a lot of technological experience and equipment and, therefore, whenever they can choose to purchase online services, because it allows them to discover and explore options (Cavagnaro et al. 2018; Monaco 2018)

The mean values of the variables under study can be found on Table 2. Regarding the variable Convenience, individuals are relatively indifferent (M = 3.7) when compared to the variable Online Experience (M = 3.93), due to the preference for using internet research. Generation Z individuals show a great degree of importance with the satisfaction with the purchase, the after-sales assistance, as well as the possibility of re-purchasing products/services online.

**Table 2.** Mean values of the variables under study.

|  | Average |
|---|---|
| Online Offer of Products/Services | 3.96 |
| Convenience | 3.7 |
| Online Experience | 3.93 |
| WOM/EWOM | 4.03 |
| Reviews on Tourism Websites | 3.7 |
| Tourism Advertising | 3.52 |
| Social networks | 3.61 |
| Tourism Websites | 4.12 |
| Trust | 4.23 |
| Price | 4.15 |
| Booking cancellation | 3.53 |

The opinion/advertising made by family and friends, due to the average values obtained (M = 4.03), shows a relatively high degree of importance. This is because they value comments and seek the best opinions, in the online experience, as we have already verified in our theoretical review (Berkup 2014).

This degree of importance can also be seen in Tourism Websites (M = 4.12), Trust (M = 4.23) and Price (M = 4.15). Factors such as technical characteristics related to online shopping, reliability with data protection and payment, and the offer of prices in online tourism when compared to tourism in physical stores, are elements of great importance for individuals of Generation Z and that motivates them to choose this method (Posner 2015).

Comments and Opinion Leaders (Reviews on Tourism Websites) have lower average values (M = 3.7), as well as Advertising in Tourism (M = 3.52), Social Networks (M = 3.61) and Cancellation Reserve (M = 3.53). These values are influenced by the respondents, with respect to comments/opinions in tourism forums and made by important opinion leaders, or important online and offline communication campaigns, although young people tend to seek the best opinions. Lower average values were observed in Images and videos, opinion of strangers (Social Networks), Cancellation and Booking. The respondents have attributed some degree of importance to these variables, however, with a lower score, and it is because of being less open to different opinions (Bejtkovský 2016).

The test for the normality of the variables showed *p*-values below the level of significance of 5%, however, due to the frequency histograms, the variables present a curve, not very far from normality.

There is a significant positive association obtained in the Spearman correlation matrix (Spearman's $\rho > 0.4$; *p*-value < 0.01), as can be seen on Table 3.

The variable Cancellation does not present a significant degree of association with any of the remaining variables, as well as the WOM/EWOM variable. Despite having weak correlations, they are positive ($\rho > 0$), however, don't allow us to identify a trend pattern when related to others.

Trust has a strong positive correlation with the variable Websites of Tourism ($\rho = 0.762$; *p*-value < 0.01) and is the highest value among the pairs of variables, revealing explanatory power to the valuation of the respondents. Individuals with greater Trust in purchasing online tourism products/services are the ones who use/visit tourism websites. Other significant correlations were obtained with the Price and Tourism Websites ($\rho = 0.548$; *p*-value < 0.01). Individuals associate good prices with the purchase of tourism products/services on websites. The conclusion is similar when we analyze the convenience of offering products and services online ($\rho = 0.530$; *p*-value < 0.01), explaining the use of tourism platforms as a practical way of purchasing products. They do it because it is more convenient.

The Spearman correlation matrix shows moderate positive association values for the Online Experience and Convenience ($\rho = 0.437$; *p*-value < 0.01) and Online Products and Services Offer ($\rho = 0.453$; *p*-value < 0.01). Advertising in Tourism and Reviews on Tourism Websites ($\rho = 0.481$; *p*-value < 0.01), Social Networks and Reviews on Tourism Websites ($\rho = 0.480$; *p*-value < 0.01) and with Advertising in Tourism ($\rho = 0.419$; *p*-value < 0.01). Tourism Websites with the variables Convenience ($\rho = 0.419$; *p*-value < 0.01), Online Products and Services Offering ($\rho = 0.478$; *p*-value < 0.01), Online Experience ($\rho = 0.502$; *p*-value < 0.01). Trust and Convenience ($\rho = 0.427$; *p*-value < 0.01), Trust and Offer of Online Products and Services ($\rho = 0.436$; *p*-value < 0.01), Trust and Online Experience ($\rho = 0.405$; *p*-value < 0.01). Finally, the Price also appears positively correlated with the variables Convenience ($\rho = 0.421$; *p*-value < 0.01), Offer of Products and Online Services ($\rho = 0.508$; *p*-value < 0.01), Online Experience ($\rho = 0.522$; *p*-value < 0.01) and Trust ($\rho = 0.438$; *p*-value < 0.01).

Analyzing the correlations with values above 0.4, we could observe that the variable Trust is associated with attributes related to the offer and availability of products/services on tourism websites, which in turn is a more convenient source of information. The experience, also associated with Trust, is an extra motivation, considering the importance of the variable Price.

These values lead to a set of characteristics associated with a personal behavior in the purchase of tourism products/services and less based on the opinions exposed on social networks or others. As we have seen, Generation Z individuals like to discover new things on the internet as well as to share their opinion about online acquisitions (Posner 2015). As they are not so influenced by traditional marketing (Mangles 2017), they always opt for online media. It is an age group with tastes very much directed to the online world, due to its technological experience (Flippin 2017; Priporas et al. 2017). This is due to the fact that the variable associated with comments, WOM/EWOM, is weakly correlated with all the others. Additionally, Cancellation is not correlated with any of the variables identified in this framework. All Spearman's correlations were validated, due to the nature of the variables.

**Table 3.** Mean values of the variables under study.

| Variable | 1 | 2 | 3 | 4 | 5 | 6 | 7 | 8 | 9 | 10 | 11 |
|---|---|---|---|---|---|---|---|---|---|---|---|
| **1 Online Products and Services Offering** | 1 | | | | | | | | | | |
| **2 Convenience** | 0.530 ** | 1 | | | | | | | | | |
| **3 Online Experience** | 0.437 ** | 0.453 ** | 1 | | | | | | | | |
| **4 WOM/EWOM** | 0.086 | 0.136 * | 0.219 ** | 1 | | | | | | | |
| **5 Reviews on Tourism Websites** | 0.129 * | 0.203 ** | 0.286 ** | 0.356 ** | 1 | | | | | | |
| **6 Tourism Advertising** | 0.027 | 0.005 | 0.212 ** | 0.198 ** | 0.481 ** | 1 | | | | | |
| **7 Social networks** | 0.169 ** | 0.186** | 0.221 ** | 0.242 ** | 0.480 ** | 0.419 ** | 1 | | | | |
| **8 Tourism Websites** | 0.419 ** | 0.478 ** | 0.502 ** | 0.276 ** | 0.278 ** | 0.189 ** | 0.282 ** | 1 | | | |
| **9 Trust** | 0.427 ** | 0.436 ** | 0.405 ** | 0.191 ** | 0.206 ** | 0.130* | 0.196 ** | 0.762 ** | 1 | | |
| **10 Price** | 0.421 ** | 0.508 ** | 0.522 ** | 0.224 ** | 0.351 ** | 0.196 ** | 0.318 ** | 0.548 ** | 0.438 ** | 1 | |
| **11 Booking cancellation** | 0.271 ** | 0.312 ** | 0.352 ** | 0.167 * | 0.198 ** | 0.149 * | 0.190 ** | 0.324 ** | 0.284 ** | 0.341 ** | 1 |

* Significant at level 0.05 (2 extremes); ** Significant at level 0.01 (2 extremes).

## 5. Conclusions and Further Recommendations

Our findings have confirmed a large part of Generation Z individuals, dependent on technology and internet use (Gollo et al. 2019), uses the internet to research and purchase tourist product/services. Our sample consists of a set of individuals with a higher degree of education, 66% of whom have already started their professional careers. The results of our research have confirmed the findings of the theoretical review, as the majority of Generation Z individuals make a tourist trip at least once a year (Nielsen 2019). We have found that the preference for searching information falls clearly on aggregating websites (Booking, Momondo, among others) by around 71.6% of respondents, and in a much smaller percentage, on travel agency websites, in 14.6% of respondents. Our research allowed us to identify the key factors in the decision of purchasing tourism products/services by Generation Z. They are Trust, Price, the use of aggregating websites, WOM/EWOM, the offer of products/services online and the Online Experience. In the opposite direction, less importance was given to Convenience, Reviews on tourism websites, Advertising, Canceling the reservation and reviews/opinions on Social Networks, this last one in contrast with the opinion of some authors (Berkup 2014; Monaco 2018; Neto et al. 2010). It was possible to identify that the variables with a stronger association with each other are Trust, price, Online Experience, the offer of products/services and the use of aggregating websites. In contrast, there was no association between the variables Cancellation, WOM/EWOM and the other variables of our framework. Generation Z individuals uses online commerce due to the trust in the service, the easy access to all information related to products/services and the possibility of comparing prices, to support their buying decisions. The use of online platforms is also a result of a positive experience when using online technologies, which are required to this generation. Digital Natives value the satisfaction resulting from a previous purchase and good customer assistance, in case of any issues during the experience. This generational group tends to trust companies invested in their online presence, yet they worry about payment security and the protection of their personal data. They also highlight Trust as a key attribute of influence when choosing a tourism website. Generation Z natives use online commerce to buy touristic products/services, due to the diversity of payment options. Despite this generation living "inside the internet" and on social networks, our study showed a relative indifference regarding the opinions and comments related to tourism, published on their social feeds. This generation may not be so influenced by the contents on the internet, as we would expect and previously identified on the theoretical review (Berkup 2014; Neto et al. 2010). However, this aspect must be better studied and confirmed with further investigation. The study of Generation Z consumption behaviors has an inherent limitation due to the age of the individuals, once a considerable portion is under 18 years old and has relatively limited decision-making power. In our study, the average age of respondents was 22 years, however, given that this generation already represents a considerable share of consumers, understanding their behaviors is quite relevant and challenging. As such, future studies should seek to understand the differences in consumption habits by Generation Z individuals, and other generations, like the precedent Generation Y, for example, in similarity to this work. Further research should also aim to clarify the interactions between all the relevant factors in online commerce, identified in this study.

This research has confirmed the consumer behavior of Generation Z individuals, previously characterized in the theoretical background. We have also confirmed this same behavior, specifically when these individuals search and buy touristic products/services on e-commerce. This study has also identified the most important factors, critical on the purchase decision process, on Generation Z consumers. This generation is already one of the most important sets of consumers of touristic products/services. In the upcoming years, Generation Z will become the most relevant consumer group, replacing older generations. Our conclusions can, therefore, be of great importance to all operators in the tourism sector because they allow for the anticipation and adaptation of marketing strategies and better management of communication channels, according to the most relevant consumer needs and behavior patterns, as identified in this study.

The main aim of our research was the tourism sector. Since Generation Z individuals are digital natives and will become the predominant generation of consumers in the short term, we can expect that the conclusions of our study and the factors identified, will be of great relevance when establishing future marketing strategies in e-commerce, as well.

**Author Contributions:** J.V. made the literature review, carried out the research and the empirical study, contributed to the discussion of the results and reviewed the manuscript. R.F. designed the methodology, contributed to the discussion of the results and reviewed the manuscript. R.A. processed the research data, performed the statistical analysis and contributed to the discussion of the results. I.P. carried out the empirical study. F.M. contributed to the discussion of the results and reviewed the manuscript. All authors have read and agreed to the published version of the manuscript.

**Funding:** This research received no external funding.

**Conflicts of Interest:** The authors declare no conflict of interest.

## Appendix A

| Variable | Categories |
|---|---|
| **How often do you travel for tourism?** | less than once a year, once a year, twice a year, three times a year, four or more times a year |
| **What is the source of digital information on tourism that you most often use for research?** | travel agency websites, aggregating websites, social networks, specialized blogs and specialized groups/pages on tourism |
| **It makes no sense to purchase tourism products/services from a travel agency physical store when I can do it from home.** | strongly disagree, disagree, neither agree nor disagree, agree, strongly agree |
| **My tourism related purchases are made online because of convenience (comfort; time efficiency)** | strongly disagree, disagree, neither agree nor disagree, agree, strongly agree |
| **When buying tourism products/services online there is a greater choice when compared to physical agencies.** | strongly disagree, disagree, neither agree nor disagree, agree, strongly agree |
| **When buying tourism products/services online you can compare different offers (prices; destinations; transport).** | strongly disagree, disagree, neither agree nor disagree, agree, strongly agree |
| **I will buy tourism products/services online again if I am satisfied with my previous purchase.** | strongly disagree, disagree, neither agree nor disagree, agree, strongly agree |
| **I would buy tourism products/services online again if I received good travel assistance in my previous purchase.** | strongly disagree, disagree, neither agree nor disagree, agree, strongly agree |
| **The more I use the internet, the more I feel inclined to purchase tourism products/services online.** | strongly disagree, disagree, neither agree nor disagree, agree, strongly agree |
| **I consider the Opinions of Relatives important.** | strongly disagree, disagree, neither agree nor disagree, agree, strongly agree |
| **I consider the Opinions of Friends important.** | strongly disagree, disagree, neither agree nor disagree, agree, strongly agree |
| **I find comments on websites and tourism forums to be important.** | strongly disagree, disagree, neither agree nor disagree, agree, strongly agree |
| **I consider Opinion Leaders (Bloggers; Instagramers; Youtubers; Vloggers) important.** | strongly disagree, disagree, neither agree nor disagree, agree, strongly agree |
| **I consider the opinions of strangers on social networks important.** | strongly disagree, disagree, neither agree nor disagree, agree, strongly agree |

| Variable | Categories |
| --- | --- |
| **Images and videos I find on social networks about tourist destinations are important.** | strongly disagree, disagree, neither agree nor disagree, agree, strongly agree |
| **Ease of use of the website.** | strongly disagree, disagree, neither agree nor disagree, agree, strongly agree |
| **Customer support.** | strongly disagree, disagree, neither agree nor disagree, agree, strongly agree |
| **Quality of information.** | strongly disagree, disagree, neither agree nor disagree, agree, strongly agree |
| **Availability (24 h a day).** | strongly disagree, disagree, neither agree nor disagree, agree, strongly agree |
| **Diversity of payment options (PayPal, visa, Mastercard and others).** | strongly disagree, disagree, neither agree nor disagree, agree, strongly agree |
| **Trust in the website.** | strongly disagree, disagree, neither agree nor disagree, agree, strongly agree |
| **Protection of personal data.** | strongly disagree, disagree, neither agree nor disagree, agree, strongly agree |
| **Payment security.** | strongly disagree, disagree, neither agree nor disagree, agree, strongly agree |
| **The internet makes it easier to compare travel prices and tourist stays.** | strongly disagree, disagree, neither agree nor disagree, agree, strongly agree |
| **The promotions offered in online tourism influence my purchase intention.** | strongly disagree, disagree, neither agree nor disagree, agree, strongly agree |
| **The price level of online tourism compared to physical agencies, influences my purchase intention.** | strongly disagree, disagree, neither agree nor disagree, agree, strongly agree |
| **On the internet, it is easier to cancel a tour booking than on a physical agency.** | strongly disagree, disagree, neither agree nor disagree, agree, strongly agree |
| **The possibility of canceling my reservation (transport and/or accommodation) online increases my purchase intention.** | strongly disagree, disagree, neither agree nor disagree, agree, strongly agree |
| **The possibility to cancel my reservation (transport or accommodation) online anticipates my purchase intention.** | strongly disagree, disagree, neither agree nor disagree, agree, strongly agree |
| **District of Residence** | Santarém, Lisbon, Leiria, Setúbal, Aveiro, Oporto, Viana do Castelo, Castelo Branco, Faro, Bragança, Évora, Coimbra, Braga, Guarda, Vila Real, Viseu, Portalegre, Azores, Madeira |
| **Respondent age (grouped)** | 10-15, 16-20, 21-25 |
| **Gender** | female, male |
| **Education degree** | basic education, high school, university education, postgraduate training |
| **Predominant Professional Situation** | student, student worker, self-employment, self-employment, unemployed |

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
