# Peer review of "Generation Z and Key-Factors on E-Commerce: A Study on the Portuguese Tourism Sector"

_admsci, doi:10.3390/admsci10040103_

Round 1
Reviewer 1 Report
"Generation Z and key-factors on e-commerce. A study on the Portuguese tourism sector"
1. INTRODUCTION:P1.L22.Could more text on the introduction be added?. Comment. Please, clarify why your paper is important. What are you going to discover? Why is this topic important? Who is going to earn more with your results about tourism, Generation Z and e-Commerce;?.It took quite a bit of reading before this point was perceived/understood. It should be clearly stated at the end of the introduction and then also prominently reflected in the findings, the discussion, and the conclusion.
2. Theoretical background. P1.L35.
Clarify and better highlight the findings that are unique to this study and not those that simply verify previous studies.
There also seems to be quite a bit of redundancy which made it difficult to easily grasp what findings were unique and critical to the entire study and the stream of Generation Z literature, in general. For example: P2.L62." Generation Z can be defined by a set of features. The main ones are the complete trust in technologies, an open mindset, they are enthusiastic, innovative, intelligent, entrepreneurial, defenders of the ethical and deontological principles. Generation Z are highly qualified, technologically experienced, innovative and creative individuals (Flippin, 2017; Priporas, Stylos & 65 Fotiadis, 2017)."
3. Research Methodology and Sample.P6.L254
In the methodology-section, I have some more concerns. First, you are lacking a clear description of and theoretical arguments for your sampling strategy.
Second, it is not clear whether the sample items were pre-tested through a pilot sample.
P6.L270: "In the present study, 233 individuals of both sexes, resident in Portugal, participated in the study and made at least one purchase of tourism through online mechanisms. As a criterion for participating in the survey, it was defined that the ages would be between 10 years and 25 years, given that it is the age group covered by generation Z. The survey was validated with the previous application to a group of three people, and through their answers and interpretation difficulties, the final version was adjusted. The application of the survey and the consequent collection of information took place from May 25 to June 11, 2020." You do not explain this methodology nor how your research project exemplifies this methodology. There is no mention of how data collection and data analysis was exhaustive (it continued) until no new information was forthcoming. Also, no mention of how one part of the data informed subsequent parts of the data collection and analysis. These are both important aspects of the grounded theory methodology.
4. Research Results and Discussion (P6-L288)
Very limited discussion section relating to the literature. Is there a relationship between the literature review and the results section of this paper?
Are there any discrepancies on result?
P9.L360. "These values lead to a set of characteristics associated with a personal behaviour in the purchase of tourism products/services and less based on the opinions exposed on social networks or other. "Why?
More explanations are necessary.
5. Conclusions and Further Recommendations:P9.L365. The conclusions to be less routine and more interesting and relevant to your findings. For example: P10.L389" Generation Z uses online commerce to buy touristic products/services, given the availability and diversity of payment options. Despite this generation living literally “inside the internet” and on social networks, our study showed a relative indifference regarding the opinions and comments about tourism, published on their social networks." What are the implications to the e-Commerce and tourism industries? There is nothing here to inspire future research or implications for practice.
6. Language: The language could be improved. Sentences too long
Author Response
Dear Sir,
As per your request, we've reviewed all the text and applied major changes to all chapters, in particular, we have tried to follow your suggestions. In the following lines, we present the detail of the applied changes and a new version of the manuscript with all changes in the attachment. Thank you for your support.
- INTRODUCTION: P1.L22.Could more text on the introduction be added?. Comment. Please, clarify why your paper is important. What are you going to discover? Why is this topic important? Who is going to earn more with your results about tourism, Generation Z and e-Commerce;?.It took quite a bit of reading before this point was perceived/understood. It should be clearly stated at the end of the introduction and then also prominently reflected in the findings, the discussion, and the conclusion.
Response 1: We have re-written the complete chapter. Please see lines 22 to 44.
- Theoretical background. P1.L35.
Clarify and better highlight the findings that are unique to this study and not those that simply verify previous studies.
There also seems to be quite a bit of redundancy which made it difficult to easily grasp what findings were unique and critical to the entire study and the stream of Generation Z literature, in general. For example: P2.L62." Generation Z can be defined by a set of features. The main ones are the complete trust in technologies, an open mindset, they are enthusiastic, innovative, intelligent, entrepreneurial, defenders of the ethical and deontological principles. Generation Z are highly qualified, technologically experienced, innovative and creative individuals (Flippin, 2017; Priporas, Stylos & 65 Fotiadis, 2017)."
Response 2: We have re-written most of the chapter. Kindly propose you to read all the chapter, in particular the yellow lines, from 47 to 49, from lines 75 to 70, from 125 to 154
- Research Methodology and Sample.P6.L254
In the methodology-section, I have some more concerns. First, you are lacking a clear description of and theoretical arguments for your sampling strategy.
Second, it is not clear whether the sample items were pre-tested through a pilot sample.
P6.L270: "In the present study, 233 individuals of both sexes, resident in Portugal, participated in the study and made at least one purchase of tourism through online mechanisms. As a criterion for participating in the survey, it was defined that the ages would be between 10 years and 25 years, given that it is the age group covered by generation Z. The survey was validated with the previous application to a group of three people, and through their answers and interpretation difficulties, the final version was adjusted. The application of the survey and the consequent collection of information took place from May 25 to June 11, 2020." You do not explain this methodology nor how your research project exemplifies this methodology. There is no mention of how data collection and data analysis was exhaustive (it continued) until no new information was forthcoming. Also, no mention of how one part of the data informed subsequent parts of the data collection and analysis. These are both important aspects of the grounded theory methodology.
Response 3: We have made substantial changes to this chapter, and added more information in order to answer to your concerns, in particular lines from 325 to 343.
- Research Results and Discussion (P6-L288)
Very limited discussion section relating to the literature. Is there a relationship between the literature review and the results section of this paper?
Are there any discrepancies on result?
P9.L360. "These values lead to a set of characteristics associated with a personal behaviour in the purchase of tourism products/services and less based on the opinions exposed on social networks or other. "Why?
More explanations are necessary.
Response 4: Most of this chapter was re-written, in particular the yellow lines, from 373 to 401, 434 to 440.
- Conclusions and Further Recommendations:P9.L365.
The conclusions to be less routine and more interesting and relevant to your findings. For example: P10.L389" Generation Z uses online commerce to buy touristic products/services, given the availability and diversity of payment options. Despite this generation living literally “inside the internet” and on social networks, our study showed a relative indifference regarding the opinions and comments about tourism, published on their social networks." What are the implications to the e-Commerce and tourism industries? There is nothing here to inspire future research or implications for practice.
Response 5: We have re-written most of the chapter. Kindly consider reading in particular lines 456 to 458, 473 to 475 and 485 to 494.
- Language: The language could be improved. Sentences too long
Response 6: We have reviewed all manuscript and have improved the quality of our “English”, I guess…

Reviewer 2 Report
The authors sufficiently summarized their objectives and approach in the abstract. The introduction and theoretical background define generation Z and lead to identifying the main variables to study.
Generally, it is an interesting view of the behaviour of generation Z.
The methodology needs some completion. The authors should precise the way of their sampling - what way have they made their choice of respondents.
The authors should summarize their theoretical contribution to the knowledge of the behaviour of the generation Z - if there is something to change when surveying this generation, for instance, or if there are some new patterns, etc.
They did not present any practical (managerial) implications for tourism practitioners/business.
The authors are to deal with some typos or other small mistakes in the text, like commas when using the word "however."
The word sex/sexes used in the text should be replaced by "gender", or even the authors could reformulate the whole phrase in the following way: "233 male and female individuals", to avoid possible conflicts with the recently delicate topic of gender.
Author Response
Dear reviewer
As per your kind request, we have reviewed all manuscript, re-written some parts and reviewed our "English". All the major changes are in colour yellow in the reviewed version of the manuscript (in the attachment). In the bellow lines please find a brief explanation on the topics you have mentioned.
Thank you
Response to Reviewer 2 Comments
The authors sufficiently summarized their objectives and approach in the abstract. The introduction and theoretical background define generation Z and lead to identifying the main variables to study.
Generally, it is an interesting view of the behaviour of generation Z.
- The methodology needs some completion. The authors should precise the way of their sampling - what way have they made their choice of respondents.
Response 1: We have re-written part of the chapter. See lines 325 to 343
- The authors should summarize their theoretical contribution to the knowledge of the behaviour of the generation Z - if there is something to change when surveying this generation, for instance, or if there are some new patterns, etc. They did not present any practical (managerial) implications for tourism practitioners/business.
Response 2: We have applied major changes to almost all chapters of the manuscript. In order to answer to your request we have re-written the chapter one, lines 23 to 44, most part of chapter five, lines 456 to 458, 474, 475, and added the lines 485 to 494
- The authors are to deal with some typos or other small mistakes in the text, like commas when using the word "however."
Response 3: We have reviewed all manuscript and have improved the quality of our “English”, I guess…
- The word sex/sexes used in the text should be replaced by "gender", or even the authors could reformulate the whole phrase in the following way: "233 male and female individuals", to avoid possible conflicts with the recently delicate topic of gender.
Response 4: Done, thank you.

Round 2
Reviewer 1 Report
Dear Author/s:
I thank the authors for this revision work, which satisfies most of my concerns about the paper "Generation Z and key-factors on e-commerce. A study on the Portuguese tourism sector"
Originality:Now OK.
Introducction: Now it is correct
Materials and methods:Now OK.
Results:Now OK.
Discussion:Now OK.
Conclusions/ Implications: Ok, but suggestions for future research: additional managerial implications in line with the findings of the study about news key-factors on e-commerce.
Language: This contains several typographical errors that are distracting, thus affecting the overall readability of the paper. A proof checking is needed for the manuscript and the references list needs to be updated with the recent evidence.
After reviewing the manuscript, congratulations on your accomplishments and keep up with the good work!
Best Regards
Author Response
Dear reviewer First of all, let us thank you again for help and support. We've followed all your suggestions, as you can find in the details below and on the attached revision (in yellow).Conclusions/ Implications: Ok, but suggestions for future research: additional managerial implications in line with the findings of the study about news key-factors on e-commerce.
Comments: We've updated the conclusions with a new phrase.
Language: This contains several typographical errors that are distracting, thus affecting the overall readability of the paper. A proof checking is needed for the manuscript and the references list needs to be updated with the recent evidence.
Comment: we have made a full document revision in order to improve the readability of the paper. We think now is much better.
After reviewing the manuscript, congratulations on your accomplishments and keep up with the good work!
Comment: Again Thank you very much!!!